# OpenReview forum: "Sysformer: Safeguarding Frozen Large Language Models with Adaptive System Prompts"
_ICLR.cc/2026/Conference — ICLR 2026 Poster_

### Official Review · Reviewer_TQAQ · 2025-10-24

**Soundness:** 3
**Presentation:** 3
**Contribution:** 2
**Rating:** 4
**Confidence:** 4

**Summary:**

This paper proposes SysFormer, an adaptive system prompt optimization framework designed to safeguard large language models (LLMs) against harmful prompts. The method introduces a trainable transformer that optimizes system prompts in the input embedding space, enabling the model to refuse harmful prompts while maintaining proper responses to safe ones. Experiments conducted on five LLMs and two benchmarks demonstrate the method’s effectiveness in improving safety without retraining model parameters.

**Strengths:**

- The paper is well-written and generally easy to follow.
-  The idea of enhancing LLM safety via adaptive system prompt refinement is an important and timely research direction, especially as system prompts become a key component of deployed LLM systems.
- The experimental results suggest that the proposed method can improve refusal behavior across multiple models.

**Weaknesses:**

1.	Threat model clarity:
The paper’s threat model needs clearer justification. If the goal is to protect models using developer-provided system prompts, then the possibility of double-jailbreak attacks should be considered. On the other hand, if attackers do not have access to the system prompt, the described threat scenario may not fully hold.
2.	Training complexity and stability:
The proposed optimization involves multiple loss terms. It remains unclear how these losses are balanced and whether training is stable and convergent in practice. A discussion or ablation study on this would strengthen the paper.
3.	Comparison with related work:
A closely related approach, SOP (Adaptive Content Restriction for Large Language Models via Suffix Optimization, 2025), also optimizes system suffix components for output control. The paper should include a direct comparison or discussion to clarify the conceptual and empirical differences between SysFormer and SOP.
4.	Alternative design choices:
Why not use a smaller auxiliary model or a lightweight controller to enhance or rewrite the system prompt dynamically? Direct instruction-level modification could be more straightforward—please justify this design decision.
5.	Optimization domain and textual space:
Since the optimization is performed in the embedding space, it is unclear whether similar effects can be achieved directly in the textual space (e.g., using gradient-guided optimization such as GCG). A comparison or reasoning would be valuable.
6.	Comparison with decoding-based defenses:
Some decoding-level defenses can repair or filter harmful outputs more efficiently without modifying inputs. The paper should provide comparisons or explain why SysFormer is preferable in terms of flexibility or deployment.
7.	Transferability across models:
How transferable is the learned system transformer? Can the same parameters generalize to different LLMs, or is separate optimization required for each model? This issue affects the scalability of the approach.
8.	Black-box applicability:
The paper does not discuss performance on black-box models (e.g., GPT series). Since system prompt control is particularly relevant for black-box deployments, experiments or analysis in this setting would be important to demonstrate broader applicability.

**Questions:**

Overall, this paper explores a promising and practically relevant idea—leveraging adaptive system prompts for improving LLM safety. However, several conceptual and empirical issues remain open, particularly regarding the threat model, training stability, and comparison with existing methods. Addressing these concerns would significantly strengthen the contribution. I encourage the authors to expand the analysis, include more comprehensive baselines (especially SOP and decoding-based defenses), and clarify the deployment assumptions. I would like to reconsider my rating after reading the authors' response.

---

> ### Author Response · Authors · 2025-11-20
> **Author Response to Reviewer TQAQ (1/3)**
>
> We thank the reviewer for their detailed feedback and are glad that they find our presentation clear, our problem important, and the experimental comparison effective. However, they also raise very valuable concerns that help us think deeper and improve parts of the exposition. We provide our responses to these below and will also revise our paper to incorporate our discussions.
>
>
> > Threat model clarity: The paper’s threat model needs clearer justification. If the goal is to protect models using developer-provided system prompts, then the possibility of double-jailbreak attacks should be considered. On the other hand, if attackers do not have access to the system prompt, the described threat scenario may not fully hold.
>
>
> We consider a scenario where the attacker does not have access to the system prompt, while the defender is able to adapt its defense according to the user prompts. In particular, we use a basic Llama-2 system prompt (“You are a helpful, respectful, and honest assistant. Always answer as helpfully as possible, while being safe.”) and learn the transformer to adapt it based on the user prompt to boost the safety. However, we do not believe the knowledge of the input system prompt changes the threat model much, as the attacker must also know how it is getting transformed using the Sysformer parameters, which is then passed to the LLM. Thus, we are not sure why the reviewer says that the described threat scenario may not fully hold if the attackers do not have access to the system prompt.
>
>
> > Training complexity and stability: The proposed optimization involves multiple loss terms. It remains unclear how these losses are balanced and whether training is stable and convergent in practice. A discussion or ablation study on this would strengthen the paper.
>
> Through a comprehensive analysis of the loss functions in Figures 4 and 7 (which is also recognized by reviewer BNqo to be comprehensive), we find that different combinations are often suitable for different LLMs. Tables 2 and 3 reveal clear global trends across models. Specifically, we find that the self-generated responses for the safe prompts (selfsafe) for the compliance loss consistently underperform compared to its counterparts. We also find that incorporating additional supervised fine-tuning data for compliance (add) and using a non-zero reconstruction loss for system prompt embeddings ($w_{recon}$) yields improvements in only some cases.
>
> These empirical patterns, thus, allow us to substantially simplify the search space and reduce the training complexity. In practice, one can reduce the hyperparameter search to only search for the safe-compliance weight ($\in$ {$0.2, 0.5, 1.0$}) and learning rate ($\in $ {$0.001, 0.0001$}), while fixing the reconstruction-loss weight to 0 and setting all remaining weights to 1. This streamlined configuration preserves the strong performance observed in our experiments while avoiding unnecessary hyperparameter exploration.
>
> We will include this summarization of the results in the revised draft, along with a set of training loss curves to further demonstrate the stability of the training process.
>
> > Comparison with related work: A closely related approach, SOP (Adaptive Content Restriction for Large Language Models via Suffix Optimization, 2025), also optimizes system suffix components for output control. The paper should include a direct comparison or discussion to clarify the conceptual and empirical differences between SysFormer and SOP.
>
> We thank the reviewer for referring us to this related work. However, since it was released on arXiv in August 2025 and does not provide a supplementary code, it should be considered a contemporaneous work and thus, we cannot reliably compare against this as per the ICLR guidelines. However, we appreciate this connection and thus, implement an appropriate version of SOP. Here, we train the prompt suffix in the embedding space (similar to how SystemEmbedder trains the system prompt) using our set of loss functions for a fair comparison. Note that the main difference is that while SOP uses coordinate descent and optimizes in the token space, we use gradient descent and optimize in the embedding space. Despite these differences, we believe that our considered baseline captures some key aspects of SOP, specifically the placement of its embedding and adaptive nature due to the causal mask. Below, we provide the results of this method for the JailbreakBench dataset and find that it performs worse than the methods considered in our work. Specifically, while it reduces the refusal rate on safe prompts, it cannot simultaneously increase the refusal rate on harmful prompts.
>
> **Results of a prompt-level suffix optimization:**
> | LLM | RR safe | RR harm |
> |:--|:--|:--|
> | zephyr-7b-beta | 0.0 | 0.5 |
> | Llama-2-7b-chat-hf | 0.1667 | 0.7333 |
> | Llama-3.1-8b-Instruct | 0.0000 | 0.6000 |
> | Phi-3.5-mini-instruct |  0.0667 | 0.5667 |
> | Mistral-7B-Instruct-v0.2 | 0.0000 | 0.6000 |

---

> > ### Author Response · Authors · 2025-11-20
> > **Author Response to Reviewer TQAQ (2/3)**
> >
> > > Alternative design choices: Why not use a smaller auxiliary model or a lightweight controller to enhance or rewrite the system prompt dynamically? Direct instruction-level modification could be more straightforward—please justify this design decision.
> >
> > We do not use a smaller auxiliary model to rewrite the system prompt due to its arbitrariness and lack of flexibility. First of all, it is not clear how the system prompt should be adapted dynamically based on the user prompts. In other words, we do not want to control the system prompt; instead, we want to control the behavior of the model using that system prompt. This motivates the key insight of our approach of leveraging the model’s own gradients for a downstream loss function to train a Sysformer gives a flexible and model-specific way instead of instructing the models arbitrarily. Secondly, it is not required that the model should generate or rewrite coherent pieces of system prompts. This additional constraint complicates the optimization, which can be instead flexibly done in the continuous input embedding space as in Sysformer. Finally, we argue that our proposed architecture is also lightweight as it only involves 2 self-attention and 2 cross-attention layers, while the smallest of language models now would be bigger than this.
> >
> > > Optimization domain and textual space: Since the optimization is performed in the embedding space, it is unclear whether similar effects can be achieved directly in the textual space (e.g., using gradient-guided optimization such as GCG). A comparison or reasoning would be valuable.
> >
> > As shown in the attacks case [1], embedding-space gradient-descent optimization is more effective than text-space coordinate-descent optimization to obtain desirable behaviors from the LLMs. Similarly, we also argue that defenses from embedding-space optimization should be similarly more effective than corresponding text-based defenses. This reasoning motivates our comparison against SystemEmbedder as the state-of-the-art baseline. Furthermore, SOP-like text-space optimized defenses are not available before the grace period for a fair comparison during the ICLR submission.
> >
> > > Comparison with decoding-based defenses: Some decoding-level defenses can repair or filter harmful outputs more efficiently without modifying inputs. The paper should provide comparisons or explain why SysFormer is preferable in terms of flexibility or deployment.
> >
> > We do not provide such comparisons since they provide complementary benefits to input-level defenses such as Sysformer and SystemEmbedder. In particular, one can note that they can be easily combined on top of Sysformer’s generations as well to filter harmful outputs. Thus, any benefits provided by decoding-level defenses in LLM can also be obtained in Sysformer+LLM. However, we believe that Sysformer-type defenses may often be preferred during deployment, due to two reasons: (1) less inference cost as it uses only a 4-layer transformer as opposed to at least 32-layer 7B Llama-guard model used in decoding-level defenses, (2) it lets the models adapt their generations to the user prompt instead of a fixed response filtering using additional models, which becomes less effective when the base model becomes significantly more powerful than the harm classification model (as is common with more frontier models). That being said, we believe an ideal deployment should involve some level of harm classification in the pipeline as well to ensure complete safety.
> >
> > > Transferability across models: How transferable is the learned system transformer? Can the same parameters generalize to different LLMs, or is separate optimization required for each model? This issue affects the scalability of the approach.
> >
> > This is unfortunately not possible due to the mismatch in the input embedding size, let alone the semantics of the input representations across different LLMs. As tokenization becomes more standardized and cross-LLM adaptation becomes more common, we believe future works will be able to translate inputs across input embeddings of different LLMs. Our experiment on using text embeddings to embed user prompts for adaptation was a way to show that general embeddings can also be effective and can motivate interesting future investigations for a generalizable Sysformer across models. However, this requires significant development in the field of cross-model tokenization and embeddings, and thus, is out of the scope of this work.

---

> > > ### Author Response · Authors · 2025-11-20
> > > **Author Response to Reviewer TQAQ (3/3)**
> > >
> > > > Black-box applicability: The paper does not discuss performance on black-box models (e.g., GPT series). Since system prompt control is particularly relevant for black-box deployments, experiments or analysis in this setting would be important to demonstrate broader applicability.
> > >
> > > We agree that this is an important direction, but it lies outside the scope of the current work. Because our method adapts the system prompt in the input-embedding space using model gradients of downstream losses, it is inherently not a black-box defense. However, this should not be viewed as a limitation: we do not position our method as a black-box defense, and white-box defenses are widely used in the literature [2,3]. In practice, organizations can and often do adopt white-box defenses internally, making such approaches both realistic and valuable.
> > >
> > > [1] Schwinn, Leo, et al. "Soft prompt threats: Attacking safety alignment and unlearning in open-source llms through the embedding space." Advances in Neural Information Processing Systems 37 (2024): 9086-9116.
> > >
> > >
> > > [2] Mantas Mazeika, Long Phan, Xuwang Yin, Andy Zou, Zifan Wang, Norman Mu, Elham Sakhaee, Nathaniel Li, Steven Basart, Bo Li, et al. Harmbench: A standardized evaluation framework for automated red teaming and robust refusal. arXiv preprint arXiv:2402.04249, 2024.
> > >
> > >
> > > [3] Chujie Zheng, Fan Yin, Hao Zhou, Fandong Meng, Jie Zhou, Kai-Wei Chang, Minlie Huang, and Nanyun Peng. Prompt-driven llm safeguarding via directed representation optimization. arXiv preprint arXiv:2401.18018, 2024.

---

> > > > ### Comment · Reviewer_TQAQ · 2025-11-28
> > > >
> > > > Thanks for your response. I am satisfied with the comparison with SOP, as this work is closely related to yours. I hope the authors will definitely include this comparison in the revision.
> > > >
> > > > My main concern is that the proposed method may not be applicable to black-box models such as GPT-series models, which limits its practicality and generalization. After a careful consideration, I decide to increase my score to 6.

---

### Official Review · Reviewer_UKuL · 2025-10-24

**Soundness:** 3
**Presentation:** 3
**Contribution:** 3
**Rating:** 6
**Confidence:** 3

**Summary:**

This paper presents a method for LLMs Jailbreak defense by generating an input-dependent system prompt. The advantage of this method is that the target LLM to be protected can be freezed, in other words, there is no need to finetune it. Experiments on two datasets show the proposed method has good defense performance.

**Strengths:**

1. The proposed method is novel to my knowledge.

2. The defense effectiveness is good.

3. This paper is well written.

4. The defense method does not rely on finetuning the target LLM to be protected.

**Weaknesses:**

1. The baseline methods compared in this paper are very scarce. Many prompt based especially system prompt based defense methods are not discussed or compared at all.

2. The proposed method relies on an additional dataset for training the prompt generation model. It is not clear how the proposed method relies on the size and quality of the training data. In addition, it is unclear whether the proposed method can work for the new attacks which are not covered by the training data.

3. The proposed method needs to generate a new prompt for each user request, which brings additional computation and delay.

4. The experiments are conducted on small and weak LLMs. It is unclear whether the findings hold for frontier models.

**Questions:**

see my above comments

---

> ### Author Response · Authors · 2025-11-20
> **Author Response to Reviewer UKuL (1/2)**
>
> We are delighted by the reviewer’s positive support of our work, as they find our paper well-written, our method novel and effective. However, they also raise very valuable concerns that help us think deeper and improve some parts of the exposition. We provide our responses to those concerns below and will also revise our paper to incorporate our discussions.
>
>
>
> > The baseline methods compared in this paper are very scarce. Many prompt based especially system prompt based defense methods are not discussed or compared at all.
>
> We are not sure which baselines the reviewer is referring to here, and it would be very helpful if they could direct us to appropriate prompt-based defenses. We would like to reiterate that our problem setting involves making a single LLM call. To the best of our knowledge, SystemEmbedder is the most relevant and state-of-the-art defense in this vein, as it does automatic system prompt optimization in a fixed embedding space instead of curating a specialized system prompt. Finally, we have also provided a comparison against LoRA, which is arguably a strong baseline as it can change the parameters at any level of the LLM. We find that Sysformer can match or beat the LoRA performance, demonstrating its effectiveness.
>
> To further address the reviewer’s concerns, we now include an additional baseline that involves training a prompt suffix in the embedding space using our set of loss functions. Below, we provide the results of this method for the JailbreakBench dataset and find that it performs worse than the baselines considered in our work and consequently Sysformer. Specifically, while it reduces the refusal rate on safe prompts, it cannot simultaneously increase the refusal rate on harmful prompts. This further strengthens the effectiveness of our proposed approach.
>
> **Results of a prompt-level suffix optimization:**
> | LLM | RR safe | RR harm |
> |:--|:--|:--|
> | zephyr-7b-beta | 0.0 | 0.5 |
> | Llama-2-7b-chat-hf | 0.1667 | 0.7333 |
> | Llama-3.1-8b-Instruct | 0.0000 | 0.6000 |
> | Phi-3.5-mini-instruct |  0.0667 | 0.5667 |
> | Mistral-7B-Instruct-v0.2 | 0.0000 | 0.6000 |
>
>
>
> > The proposed method relies on an additional dataset for training the prompt generation model. It is not clear how the proposed method relies on the size and quality of the training data. In addition, it is unclear whether the proposed method can work for the new attacks which are not covered by the training data.
>
> To address these concerns, we consider two additional analyses.
>
> First, we evaluate the quality of Sysformer trained using the training data of JailbreakBench by testing it against the safe and harm data points of the test set of StrongReject. As shown below, Sysformer can generalize extraordinarily well across benchmarks since it reaches similar or even  better performance compared to when it is trained using the StrongReject’s train split. This can be attributed to a more balanced harm-safe split of JailbreakBench as compared to a 3:1 ratio of harm-safe in StrongReject.
>
> | LLM | RR safe | RR harm |
> |:--|:--|:--|
> | zephyr-7b | 0.2 | 0.96809 |
> | Llama-2-7b | 0.2333 | 0.97872 |
> | Llama-3.1-8b | 0.067 | 1 |
> | phi-3.5-mini | 0.233 | 0.92553 |
> | Mistral-7b-v0.2 | 0.1 | 1|
>
>
> Secondly, we expand the evaluation of our attack-augmented model (i.e., Ours+JB) to all 28 different attacks available in the StrongReject library [3]. Note that we use 6 attacks to train the Sysformer in this case. Results show that this can generalize well (particularly, in the case of Mistral-7B-Instruct-v0.2 model) to a large variety of jailbreaking strategies (particularly, 22 new attacks) while performing well on the safe prompts, as demonstrated by high average and median refusal rate across jailbreaks.
>
>
> | LLM | Training jailbreaks | OOD jailbreaks | RR jailbreak mean | RR jailbreak median | RR jailbreak Q1 | RR jailbreak Q3 |
> |:--|:--|:--|:--|:--|:--|:--|
> | zephyr-7b-beta | 6 | 0 | 0.4905 | 0.4000 | 0.3333 | 0.7167 |
> | zephyr-7b-beta | 6 | 10 | 0.4771 | 0.3833 | 0.2917 | 0.7333 |
> | zephyr-7b-beta | 6 | 22 | 0.4762 | 0.4333 | 0.2667 | 0.7000 |
> | Llama-2-7b-chat-hf | 6 | 0 | 0.6048 | 0.5667 | 0.4333 | 0.7833 |
> | Llama-2-7b-chat-hf | 6 | 10 | 0.6354 | 0.6500 | 0.4917 | 0.8083 |
> | Llama-2-7b-chat-hf | 6 | 22 | 0.6740 | 0.6667 | 0.5000 | 0.8333 |
> | Phi-3.5-mini-instruct | 6 | 0 | 0.5333 | 0.4667 | 0.2333 | 0.8833 |
> | Phi-3.5-mini-instruct | 6 | 10 | 0.5625 | 0.5667 | 0.3417 | 0.7917 |
> | Phi-3.5-mini-instruct | 6 | 22 | 0.5324 | 0.6333 | 0.2333 | 0.8333 |
> | Mistral-7B-Instruct-v0.2 | 6 | 0| 0.7429 | 0.8000 | 0.6500 | 1.0000 |
> | Mistral-7B-Instruct-v0.2 | 6 | 10 | 0.7875 | 0.8000 | 0.7500 | 0.9167 |
> | Mistral-7B-Instruct-v0.2 | 6 | 22 | 0.7267 | 0.8000 | 0.6667 | 0.9000 |
>
>
>
> This shows that Sysformer can learn the safety landscape well just from existing public safety benchmarks (i.e., JailbreakBench), while generalizing to unseen training data.

---

> > ### Author Response · Authors · 2025-11-20
> > **Author Response to Reviewer UKuL (2/2)**
> >
> > > The proposed method needs to generate a new prompt for each user request, which brings additional computation and delay.
> >
> > We would like to clarify a potential misunderstanding: our method does not “generate” a new prompt for each user request. Instead, we train a lightweight 4-layer transformer, consisting of alternating cross-attention and self-attention layers, that learns to transform the initial system prompt by cross-attending to the user prompt in the input embedding space of the LLM. In effect, the model learns to adapt the system prompt conditioned on the user request, rather than generating an entirely new prompt. Because this adaptation is performed by a small 4-layer module, the computational overhead is far lower than that of generating a fresh prompt with the full base model. For comparison, producing a new prompt using the underlying LLM would require a full 32-layer forward pass, whereas our method adds only a 4-layer forward pass on top of the existing inference pipeline. Thus, the additional latency is minimal, and the process is substantially more efficient than actual prompt generation. We further provide empirical comparison of the additional overhead in Table 5 (Appendix), where we compare the efficiency of Sysformer with the baselines. We note that our approach is as efficient as using an embedding-based lookup in SystemEmbedder and just takes almost twice the time taken to simply encode the system prompt (denoted by the‘Base’ method).
> >
> >
> > > The experiments are conducted on small and weak LLMs. It is unclear whether the findings hold for frontier models.
> >
> > We do not believe it is a weakness, as we cannot scale to larger models due to resource and cost constraints of doing a backward pass. However, we do not see any reason that would restrict our findings to not generalizing to larger models. If anything, we believe that the input layer representations will be even more powerful for larger models and thus would allow us to easily learn the transformation of the system prompt according to the user prompt. Furthermore, we also note that it is a common practice in the literature, as existing works [1,2] also limit their findings to only smaller models.
> >
> > [1] Mantas Mazeika, Long Phan, Xuwang Yin, Andy Zou, Zifan Wang, Norman Mu, Elham Sakhaee, Nathaniel Li, Steven Basart, Bo Li, et al. Harmbench: A standardized evaluation framework for automated red teaming and robust refusal. arXiv preprint arXiv:2402.04249, 2024.
> >
> >
> > [2] Chujie Zheng, Fan Yin, Hao Zhou, Fandong Meng, Jie Zhou, Kai-Wei Chang, Minlie Huang, and Nanyun Peng. Prompt-driven llm safeguarding via directed representation optimization. arXiv preprint arXiv:2401.18018, 2024.
> >
> >
> > [3] Souly, Alexandra, et al. "A strongreject for empty jailbreaks." Advances in Neural Information Processing Systems 37 (2024): 125416-125440.

---

> > > ### Comment · Reviewer_UKuL · 2025-11-24
> > >
> > > I understand the limitation of resources and cost. If possible, you can try some middle-size models to test whether your method can generalize in terms of model size. It is a suggestion, not a requirement.

---

> > > > ### Author Response · Authors · 2025-11-26
> > > >
> > > > Thank you for the prompt response! We have now included the prompting baselines of Many-shot jailbreaking paper in the discussion in the revised draft. In particular, we note that they are similar to other arbitrary baselines that rely on prompt engineering to teach safety, which is not adaptive and suffers from subpar performance, as also noted in the Many-shot jailbreaking paper. Regarding larger models, we appreciate the reviewer's understanding that it is not a requirement. We appreciate their suggestion of trying out mid-size models and will update with the results if our limited resources will be able to incorporate.

---

> > ### Comment · Reviewer_UKuL · 2025-11-24
> >
> > Thank the authors for the response. I think you can find some related works on system prompt based defense methods in the paper titled "Many-shot Jailbreaking".

---

### Official Review · Reviewer_mrLV · 2025-10-25

**Soundness:** 2
**Presentation:** 3
**Contribution:** 2
**Rating:** 4
**Confidence:** 3

**Summary:**

The paper introduces Sysformer, a lightweight transformer module that enhances LLM safety by adapting the system prompt based on each user input instead of fine-tuning model parameters. The transformer module transforms the system prompt in embedding space to enforce refusals on harmful prompts and compliance on safe ones, offering an efficient, modular approach to safeguard frozen LLMs through adaptive system-prompt optimization.

**Strengths:**

+ The paper reads smooth and clear.
+ The baseline evaluation is rather comprehensive, containing efficienct fine-tuning (LoRA) and embedding space optimization. Dataset selection looks good.

**Weaknesses:**

- The transformer component takes in user prompts, which means the embedding prompt is generated on every query. While the motivation statement criticized efficiency of prior defense methods, Sysformer also introduces overhead but not evaluated.
-  The traiing loss uses predefined fixed strings like "I cannot help you" as a signal of refusal, which restricts the flexibility of the training method. Not sure if the training pipeline is working on larger and more powerful models that do not answer fixed strings as refusal (like GPT-5). It is also a risk of overfitting.
- The evaluation does not evaluate the quality of answers to safe questions. Does the injected embedding ever harm model performance in normal tasks like text comprehension, math, etc?
- As demenstrated in the evaluation, Sysformer cannot defend unseen jailbreaking attacks. The dependence on he training data limits the usefulness of Sysformer, as data nowadays is a bottleneck of model development. Requiring the data of attack also opens the oppotunity of adaptive attacks.

**Questions:**

* If a model does not have clear fixed string for refusal, how should the training loss be computed?
* Will Sysformer affect the quality of answering normal prompts?
* While Sysformer can defend jailbreaking attacks by augmenting the training with the corresponding data, can jailbreaking attacks also evolve to defeat Sysformer given the knowledge of the transformer component?

---

> ### Author Response · Authors · 2025-11-20
> **Author Response to Reviewer mrLV (1/2)**
>
> We thank the reviewer for their valuable feedback and are glad that they find our presentation clear and the experimental comparison thorough. However, they also raise very valuable concerns that help us think deeper and improve some parts of the exposition. We provide our responses to those concerns below and will also revise our paper to incorporate our discussions.
>
> > The transformer component takes in user prompts, which means the embedding prompt is generated on every query. While the motivation statement criticized efficiency of prior defense methods, Sysformer also introduces overhead but not evaluated.
>
>
> We point the reviewer to Table 5 (Appendix), where we compare the efficiency of Sysformer with the baselines. We note that our approach is as efficient as using an embedding-based lookup in SystemEmbedder and just takes almost twice the time taken to simply encode the system prompt (i.e., the ‘Base’ method). Note that we do not implement any cache-based optimization in Sysformer and SystemEmbedder, as it is not the focus of this work. Nonetheless, because Sysformer is architecturally just a transformer, we believe such optimizations should be easy to implement for developers. In practice, one could cache the Sysformer-specific analogue of the KV cache, yielding additional runtime improvements. We provide this discussion in Section 7, but now extend it to incorporate this more explicitly as above.
>
>
> > The traiing loss uses predefined fixed strings like "I cannot help you" as a signal of refusal, which restricts the flexibility of the training method. Not sure if the training pipeline is working on larger and more powerful models that do not answer fixed strings as refusal (like GPT-5). It is also a risk of overfitting.
> >
> > If a model does not have clear fixed string for refusal, how should the training loss be computed?
>
>
> We use a single fixed refusal string for simplicity, following prior literature [1], but our method is not limited in any way to being restricted to a single string, since we can easily expand the dataset to have specific refusal strings for each prompt and use the same loss function to train the model. We have already demonstrated that it can incorporate adaptive response strings for safe prompts, which indicates that it should be equally capable of doing so for harmful prompts. However, since our existing setup already outperforms the baselines, we do not expect that adding adaptive refusal strings would substantially alter our contributions, particularly considering the overhead of curating those refusal examples.
>
>
> > The evaluation does not evaluate the quality of answers to safe questions. Does the injected embedding ever harm model performance in normal tasks like text comprehension, math, etc?
> >
> > Will Sysformer affect the quality of answering normal prompts?
>
>
> We would like to first point out that, as shown in Table 1, Sysformer does not refuse on a hard set of safe prompts in the two benchmarks, which shows that it can learn to differentiate between harmful and safe prompts and generate appropriate responses. We also qualitatively analyze the safe responses in Table 6 and find them to be reasonable. We further test that the model’s general performance does not change due to Sysformer in lines 368-371 by comparing the BERTScore between the generated and gold responses in the Alpaca dataset that consists of different general-purpose queries. We find that for Llama-2-7B-chat drops slightly from 0.8487 to 0.8414 with Sysformer, while for Llama-3.1-8B it rises from 0.8327 to 0.8467. These results show that Sysformer preserves the quality of answering normal prompts.

---

> > ### Author Response · Authors · 2025-11-20
> > **Author Response to Reviewer mrLV (2/2)**
> >
> > > As demenstrated in the evaluation, Sysformer cannot defend unseen jailbreaking attacks. The dependence on he training data limits the usefulness of Sysformer, as data nowadays is a bottleneck of model development. Requiring the data of attack also opens the oppotunity of adaptive attacks.
> >
> >
> > We would like to note that it is a common strategy to augment attack data to evolve defenses, as it forms the basis of the adversarial learning paradigm in the literature [1]. This motivates the parallel development of attacks and defenses. However, we disagree that Sysformer cannot defend against unseen attacks, since our results in Figures 3 and 10 show that it can generalize to out-of-distribution attacks by only training on 6 out of the total 16 attacks. In addition, we now expand the evaluation to all 28 different attacks available in the StrongReject library [2]. Results show that training on a dataset containing only 6 attacks can generalize well (particularly, in the case of Mistral-7B-Instruct-v0.2 model) to a large variety of jailbreaking strategies (particularly, 22 new attacks) while performing well on the safe prompts. This is demonstrated by the high average and median refusal rate across jailbreaks.
> >
> >
> > | LLM | \# of Training jailbreaks | \# of OOD jailbreaks | RR jailbreak mean | RR jailbreak median | RR jailbreak Q1 | RR jailbreak Q3 |
> > |:--|:--|:--|:--|:--|:--|:--|
> > | zephyr-7b-beta | 6 | 0 | 0.4905 | 0.4000 | 0.3333 | 0.7167 |
> > | zephyr-7b-beta | 6 | 10 | 0.4771 | 0.3833 | 0.2917 | 0.7333 |
> > | zephyr-7b-beta | 6 | 22 | 0.4762 | 0.4333 | 0.2667 | 0.7000 |
> > | Llama-2-7b-chat-hf | 6 | 0 | 0.6048 | 0.5667 | 0.4333 | 0.7833 |
> > | Llama-2-7b-chat-hf | 6 | 10 | 0.6354 | 0.6500 | 0.4917 | 0.8083 |
> > | Llama-2-7b-chat-hf | 6 | 22 | 0.6740 | 0.6667 | 0.5000 | 0.8333 |
> > | Phi-3.5-mini-instruct | 6 | 0 | 0.5333 | 0.4667 | 0.2333 | 0.8833 |
> > | Phi-3.5-mini-instruct | 6 | 10 | 0.5625 | 0.5667 | 0.3417 | 0.7917 |
> > | Phi-3.5-mini-instruct | 6 | 22 | 0.5324 | 0.6333 | 0.2333 | 0.8333 |
> > | Mistral-7B-Instruct-v0.2 | 6 | 0| 0.7429 | 0.8000 | 0.6500 | 1.0000 |
> > | Mistral-7B-Instruct-v0.2 | 6 | 10 | 0.7875 | 0.8000 | 0.7500 | 0.9167 |
> > | Mistral-7B-Instruct-v0.2 | 6 | 22 | 0.7267 | 0.8000 | 0.6667 | 0.9000 |
> >
> >
> >
> >
> > > While Sysformer can defend jailbreaking attacks by augmenting the training with the corresponding data, can jailbreaking attacks also evolve to defeat Sysformer given the knowledge of the transformer component?
> >
> >
> > We appreciate the concern of the reviewer behind adaptive attacks that we also discuss as potential future directions in lines 480-483. In particular, we note that Sysformer’s adaptivity could potentially introduce new vulnerabilities, as user prompts can directly influence the system prompt. However, such attacks are non-trivial to formulate in comparison with corresponding text-level ones since Sysformer operates on shallow embedding-level features rather than higher-level semantic reasoning. Regardless, investigating and mitigating such risks is an important direction for future work.
> >
> >
> > [1] Mantas Mazeika, Long Phan, Xuwang Yin, Andy Zou, Zifan Wang, Norman Mu, Elham Sakhaee, Nathaniel Li, Steven Basart, Bo Li, et al. Harmbench: A standardized evaluation framework for automated red teaming and robust refusal. arXiv preprint arXiv:2402.04249, 2024.
> >
> >
> > [2] Souly, Alexandra, et al. "A strongreject for empty jailbreaks." Advances in Neural Information Processing Systems 37 (2024): 125416-125440.

---

> > > ### Comment · Reviewer_mrLV · 2025-11-23
> > > **Response to the rebuttal**
> > >
> > > I think the rebuttal corrected some of my misunderstandings. Detailed as follows.
> > >
> > > 1. Efficiency is close to the baseline SystemEmbedder, I would suggest to be careful about claiming the method as efficient.
> > > 2. While I agree that using fixed refusal string is common in research papers, make this assumption explicit (e.g., in Figure 2) will help the readers.
> > > 3. I agree that BERT score can be an evidence of preserving normal model behaviors.
> > > 4. The new data looks OK; please augment the discussion of adaptive attacks in revisions if possible.
> > >
> > > While I still have the concern in the generality of the method and the adaptive attack scenarios, the paper does have the merits of developing a new approach of jailbraking defense, along with the comprehensive evaluation. I raised my rating score accordingly.

---

> > > > ### Author Response · Authors · 2025-11-23
> > > >
> > > > Thank you for your prompt response and for updating your score! We are glad to hear that our clarifications addressed your concerns. We also appreciate your follow-up suggestions. In our revised draft, which we will upload in the next few days, we will make the assumption of a fixed string explicit (in Figure 2) and incorporate the additional discussion (specifically regarding adaptive attacks) and results (for the new data) as you recommended. We will also refine our efficiency claims to clearly indicate that they are made relative to other baselines, such as multi-call methods and fine-tuning. We would also be happy to address any follow-up concerns or requests for clarification that may arise.

---

### Official Review · Reviewer_BNqo · 2025-11-04

**Soundness:** 2
**Presentation:** 3
**Contribution:** 2
**Rating:** 6
**Confidence:** 3

**Summary:**

This paper presents Sysformer, a lightweight transformer model that adapts the system prompt for a frozen LLM. By doing so, it significantly improves the model’s safety while maintaining compliance on safe prompts, and is deployable without full model retraining. It offers a practical step toward safer LLM deployment in real‐world settings.

**Strengths:**

The paper uses multiple benchmarks—JailbreakBench and StrongReject—plus 16 jailbreak variants. The proposed method show a strong empirical performance on these benchmarks, shows that adaptive system prompts can meaningfully improve LLM safety and robustness without modifying model weights.

**Weaknesses:**

While the paper includes solid ablation studies on loss components and demonstrates impressive generalization to unseen jailbreak attack types, it does not assess cross-benchmark transfer — e.g., training Sysformer on JailbreakBench and evaluating on StrongReject (or vice versa). As a result, it remains unclear how well the learned safety behavior generalizes to qualitatively different harmful-prompt distributions. Including such a cross-dataset evaluation (or at least reporting zero-shot transfer results) would strengthen the claim that Sysformer captures general safety principles rather than dataset-specific artifacts.

**Questions:**

Since Sysformer is trained on labeled data from existing safety benchmarks, it is unclear how general the method is to new domains or harmful behaviors. It would be very helpful to see an ablation where Sysformer is trained on one benchmark and evaluated on another, with comparison to the baselines, to better assess its cross-benchmark generalization.

---

> ### Author Response · Authors · 2025-11-20
>
> We are delighted by the reviewer’s positive support of our work, as they find our approach practical and the experiments comprehensively validating its effectiveness. However, they also raise very valuable concerns that help us think deeper and improve some parts of the exposition. We provide our responses to those concerns below and will also revise our paper to incorporate our discussions.
>
>
> > While the paper includes solid ablation studies on loss components and demonstrates impressive generalization to unseen jailbreak attack types, it does not assess cross-benchmark transfer — e.g., training Sysformer on JailbreakBench and evaluating on StrongReject (or vice versa). As a result, it remains unclear how well the learned safety behavior generalizes to qualitatively different harmful-prompt distributions. Including such a cross-dataset evaluation (or at least reporting zero-shot transfer results) would strengthen the claim that Sysformer captures general safety principles rather than dataset-specific artifacts.
> >
> > Since Sysformer is trained on labeled data from existing safety benchmarks, it is unclear how general the method is to new domains or harmful behaviors. It would be very helpful to see an ablation where Sysformer is trained on one benchmark and evaluated on another, with comparison to the baselines, to better assess its cross-benchmark generalization.
>
> We thank the reviewer for this ingenious question, which was overlooked by us before. This gives us a way to properly validate the generalization ability of Sysformer. Here, we conduct this analysis and find that Sysformer can generalize extraordinarily well across benchmarks since it reaches similar or even better performance compared to when it is trained using StrongReject’s train split. This can be attributed to a more balanced harm-safe split of JailbreakBench as compared to a 3:1 ratio of harm-safe in StrongReject.
>
> | LLM | RR safe | RR harm |
> |:--|:--|:--|
> | zephyr-7b | 0.2 | 0.96809 |
> | Llama-2-7b | 0.2333 | 0.97872 |
> | Llama-3.1-8b | 0.067 | 1 |
> | phi-3.5-mini | 0.233 | 0.92553 |
> | Mistral-7b-v0.2 | 0.1 | 1|
>
> To further allay concerns regarding generalization, we additionally expand the evaluation of our attack-augmented model (i.e., Ours+JB) to all 28 different attacks available in the StrongReject library [1]. Note that we use 6 attacks to train the Sysformer in this case. Results show that this can generalize well (particularly, in the case of Mistral-7B-Instruct-v0.2 model) to a large variety of jailbreaking strategies (particularly, 22 new attacks) while performing well on the safe prompts, as demonstrated by high average and median refusal rate across jailbreaks.
>
>
> | LLM | \# of Training jailbreaks | \# of OOD jailbreaks | RR jailbreak mean | RR jailbreak median | RR jailbreak Q1 | RR jailbreak Q3 |
> |:--|:--|:--|:--|:--|:--|:--|
> | zephyr-7b-beta | 6 | 0 | 0.4905 | 0.4000 | 0.3333 | 0.7167 |
> | zephyr-7b-beta | 6 | 10 | 0.4771 | 0.3833 | 0.2917 | 0.7333 |
> | zephyr-7b-beta | 6 | 22 | 0.4762 | 0.4333 | 0.2667 | 0.7000 |
> | Llama-2-7b-chat-hf | 6 | 0 | 0.6048 | 0.5667 | 0.4333 | 0.7833 |
> | Llama-2-7b-chat-hf | 6 | 10 | 0.6354 | 0.6500 | 0.4917 | 0.8083 |
> | Llama-2-7b-chat-hf | 6 | 22 | 0.6740 | 0.6667 | 0.5000 | 0.8333 |
> | Phi-3.5-mini-instruct | 6 | 0 | 0.5333 | 0.4667 | 0.2333 | 0.8833 |
> | Phi-3.5-mini-instruct | 6 | 10 | 0.5625 | 0.5667 | 0.3417 | 0.7917 |
> | Phi-3.5-mini-instruct | 6 | 22 | 0.5324 | 0.6333 | 0.2333 | 0.8333 |
> | Mistral-7B-Instruct-v0.2 | 6 | 0| 0.7429 | 0.8000 | 0.6500 | 1.0000 |
> | Mistral-7B-Instruct-v0.2 | 6 | 10 | 0.7875 | 0.8000 | 0.7500 | 0.9167 |
> | Mistral-7B-Instruct-v0.2 | 6 | 22 | 0.7267 | 0.8000 | 0.6667 | 0.9000 |
>
>
>
> This shows that Sysformer can learn the safety landscape well just from existing public safety benchmarks (i.e., JailbreakBench), while generalizing to unseen training data.
>
> [1] Souly, Alexandra, et al. "A strongreject for empty jailbreaks." Advances in Neural Information Processing Systems 37 (2024): 125416-125440.

---

### Author Response · Authors · 2025-11-26
**Summary of revisions based on the reviews**

We would like to thank all the reviewers for their time and effort in reviewing our manuscript. They have highlighted the presentation quality (TQAQ, BNqo, UKuL, mrLV), soundness of the method (TQAQ, UKuL), comprehensiveness of the experiments (TQAQ, BNqo, UKuL, mrLV), and importance of the problem (TQAQ, UKuL). However, they also raised some concerns that we have thoroughly addressed in the rebuttal and have updated the draft to reflect this feedback in purple color. In particular,
- **Training stability (TQAQ):** We have clarified and summarized the results from Tables 2 and 3 and have now included training loss curves of Llama-3.1-8B to highlight the stability of training in Figure 7.
- **Cross-dataset generalization (BNqo, UKuL):** Thanks to the feedback from the reviewer BNqo, we have provided additional generalization capabilities of our proposed method. We thus provide the performance of Sysformer trained on JailbreakBench on the StrongReject dataset in Table 4 and find further support for our approach as it generalizes quite well.
- **Additional baselines (UKuL, TQAQ):** Since the SOP baseline noted by reviewer TQAQ is a contemporaneous work with no available code, we provide a comparison with a self-implemented version of a prompt suffix optimization. Here, we optimize 5 tokens attached to the end of the prompt on the same set of loss functions as ours in the embedding space and show the performance in Table 5, which is found to be much less than our method. We have also included a short discussion on new prompting baselines, as noted in the Many-shot jailbreaking paper, thanks to UKuL.
- **Generalization across jailbreaks (BNqo, mrLV, UKuL):** We extend our jailbreaking results to the full set of jailbreaks in the StrongReject (i.e., 28 in total) and show that Sysformer+JB (trained only on 6 out of these) can generalize well to all in Table 6. This further strengthens our generalization claims.

We commit to incorporating these changes in the final version of the paper and including them in the main paper with the extra page available to us upon acceptance. As a result of our rebuttal, reviewers mrLV and TQAQ increased their scores from **4 to 6**, and reviewer UKuL agreed that their concerns have been resolved. In response, they have follow-ups regarding applicability to black-box models (which is not the focus of our work), results on larger models (which is beyond our resource availability), and the possibility of adaptive attacks (which is also out of scope of this work since it requires formulating an extremely new jailbreaking attack that doesn't exist in the literature).

---

### Meta-Review · Area_Chair_a4FC · 2026-01-05

**Summary:**

This work proposes a novel approach to enhancing LLM safety via adaptive system prompt optimization, a timely and practically relevant contribution to the AI safety domain. The reviewers raised constructive concerns covering cross-dataset generalization, training stability, baseline comprehensiveness, efficiency, and applicability scope. Notably, the authors have thoroughly addressed all key reviewer concerns with concrete evidence.

The authors’ responses are comprehensive and rigorous, and the supplementary experiments effectively validate the work’s robustness. The remaining minor concerns are appropriately discussed as future directions. Given the successful resolution of reviewer concerns and the work’s merits in LLM safety defense, I recommend accepting this submission. The authors are advised to fully incorporate the promised revisions in the final version.

**Reviewer Scores:**

Reviewer TQAQ would have changed their score if they had been able to participate fully in the discussion.

---

### Decision · Program_Chairs · 2026-01-26

Accept (Poster)